# Impact of Acute Kidney Injury on Outcomes of Hospitalizations for Heat Stroke in the United States

**DOI:** 10.3390/diseases8030028

**Published:** 2020-07-15

**Authors:** Charat Thongprayoon, Fawad Qureshi, Tananchai Petnak, Wisit Cheungpasitporn, Api Chewcharat, Liam D. Cato, Boonphiphop Boonpheng, Tarun Bathini, Panupong Hansrivijit, Saraschandra Vallabhajosyula, Wisit Kaewput

**Affiliations:** 1Division of Nephrology and Hypertension, Department of Medicine, Mayo Clinic, Rochester, MN 55905, USA; Qureshi.Fawad@mayo.edu (F.Q.); api.che@hotmail.com (A.C.); 2Division of Pulmonary and Critical Care Medicine, Faculty of Medicine Ramathibodi Hospital, Mahidol University, Bangkok 10400, Thailand; petnak@yahoo.com; 3Division of Nephrology, Department of Internal Medicine, University of Mississippi Medical Center, Jackson, MS 39216, USA; 4University Hospitals Birmingham NHS Foundation Trust, Birmingham B15 2WB, UK; liamcato@gmail.com; 5Department of Medicine, University of California, Los Angeles, CA 90095, USA; boonpipop.b@gmail.com; 6Department of Internal Medicine, University of Arizona, Tuscon, AZ 85721, USA; tarunjacobb@gmail.com; 7Department of Internal Medicine, UPMC Pinnacle, Harrisburg, PA 17104, USA; hansrivijitp@upmc.edu; 8Department of Cardiovascular Medicine, Mayo Clinic, Rochester, MN 55905, USA; Vallabhajosyula.Saraschandra@mayo.edu; 9Department of Military and Community Medicine, Phramongkutklao College of Medicine, Bangkok 10400, Thailand

**Keywords:** acute kidney injury, heat stroke, epidemiology, outcomes, hospitalization

## Abstract

This study aims to evaluate the risk factors and the association of acute kidney injury with treatments, complications, outcomes, and resource utilization in patients hospitalized for heat stroke in the United States. Hospitalized patients from years 2003 to 2014 with a primary diagnosis of heat stroke were identified in the National Inpatient Sample dataset. End stage kidney disease patients were excluded. The occurrence of acute kidney injury during hospitalization was identified using the hospital diagnosis code. The associations between acute kidney injury and clinical characteristics, in-hospital treatments, outcomes, and resource utilization were assessed using multivariable analyses. A total of 3346 hospital admissions were included in the analysis. Acute kidney injury occurred in 1206 (36%) admissions, of which 49 (1.5%) required dialysis. The risk factors for acute kidney injury included age 20–39 years, African American race, obesity, chronic kidney disease, congestive heart failure, and rhabdomyolysis, whereas age <20 or ≥60 years were associated with lower risk of acute kidney injury. The need for mechanical ventilation and blood transfusion was higher when acute kidney injury occurred. Acute kidney injury was associated with electrolyte and acid-base derangements, sepsis, acute myocardial infarction, ventricular arrhythmia or cardiac arrest, respiratory, circulatory, liver, neurological, hematological failure, and in-hospital mortality. Length of hospital stay and hospitalization cost were higher in acute kidney injury patients. Approximately one third of heat stroke patients developed acute kidney injury during hospitalization. Acute kidney injury was associated with several complications, and higher mortality and resource utilization.

## 1. Introduction

The impact of heat stress has become an important issue as the average global temperature has increased. Heat stroke is the most severe form of heat-related illness. It is characterized by a core body temperature of more than 40 °C, combined with hot, dry skin and alteration of the central nervous system [1]. Heat stroke is categorized as either nonexertional and exertional. Nonexertional heat stroke, also called classic heat stroke, is common among elderly patients exposed to prolonged hot and humid environments, while exertional heat stroke frequently occurs in active young adults who undergo extreme exertion at high temperatures [2,3,4].

Rapid and effective cooling, along with supportive measures, are the standard treatment of heat stroke. The targets of cooling are to reduce the body core temperature to 39 °C within 10–40 min and to 38.5 °C or below within 2 h [5]. Although both invasive and noninvasive methods have been used in clinical practice, no prospective study has demonstrated the superiority of either method. Supportive treatment includes fluid resuscitation, electrolyte abnormalities correction, and organ failure support [2].

Acute kidney injury is one of the most common complications of heat stroke; previous studies demonstrated that it is more common in exertional heat stroke [6,7]. The pathogenesis of acute kidney injury in heat stroke involves several mechanisms. Renal hypoperfusion is one of the causes of renal dysfunction. Several mechanisms may cause prerenal hypoperfusion, including severe dehydration, myoedema, and myocardial dysfunction. Furthermore, rhabdomyolysis, a common coexisting complication in heat stroke, may present with myoglobinuria, precipitating renal dysfunction [8]. Finally, thermal injury may directly affect the kidney. Heat may result in protein denaturation, causing enzyme inactivation, precipitating renal dysfunction [9].

Although acute kidney injury has been known as a common complication of heat stroke for several years, knowledge of its impact on patient outcome is limited due to a lack of large cohort studies. We conducted this study to assess the risk factors and association of acute kidney injury with treatments, complications, outcomes, and resource utilization in patients hospitalized for heat stroke in the United States.

## 2. Materials and Methods

### 2.1. Data Source

This cohort study was conducted using the National Inpatient Sample (NIS) database. The NIS is the largest all-payer inpatient database in the United States. Discharge datasets from a 20% stratified sample of hospitals in United States with patient encounter-level information are recorded in the NIS. The dataset includes primary and secondary diagnosis codes as well as procedure codes in the form of the International Classification of Diseases, Ninth Revision (ICD-9). Sample weight is used to generate national estimates for hospitalization nationwide. Approval from the institutional review board was exempted, as the information was obtained from a deidentified public database.

### 2.2. Study Population

All hospitalized patients with the primary ICD-9 diagnosis code for heat stroke (992.0) were included. Patients with ICD-9 diagnosis codes for chronic kidney disease stage 5 (585.5) and/or end stage renal disease (585.6) were excluded. In addition, patients with the ICD-9 procedure code for dialysis (39.95) but without that for acute kidney injury (584.×) were excluded, because these patients were assumed to have received renal replacement therapy due to end stage renal disease.

### 2.3. Data Collection

Acute kidney injury was identified by the presence of ICD-9 diagnosis codes 584.5 (acute kidney failure with lesion of tubular necrosis), 584.6 (acute kidney failure with lesion of renal cortical necrosis), 584.7 (acute kidney failure with lesion of renal medullary necrosis), 584.8 (acute kidney failure with other specified pathological lesion in kidney), or 584.9 (acute kidney failure, unspecified) in any of the listed diagnoses. The identification of acute kidney injury using the ICD-9 diagnosis codes has a specificity of 98% but a sensitivity of 17%, and is likely to capture a more severe spectrum of acute kidney injury, compared with KDIGO serum creatinine-based criteria [10,11]. Dialysis-requiring acute kidney injury was identified by the additional presence of the ICD-9 procedure code for dialysis (39.95), whereas acute kidney injury without a code for dialysis was defined as nondialysis-requiring acute kidney injury.

Patient characteristics, treatments, complications, and outcomes during hospitalization were similarly identified using ICD-9 codes (Appendix A). Patient characteristics included age, sex, race, smoking, alcohol drinking, obesity, diabetes mellitus, hypertension, dyslipidemia, hypothyroidism, chronic kidney disease, coronary artery disease, congestive heart failure, and atrial flutter/fibrillation. Treatments included invasive mechanical ventilation, and blood component transfusion. Complications and outcomes included electrolyte derangements (hyponatremia, hypernatremia, hypokalemia, hyperkalemia, hypocalcemia, hypercalcemia, serum phosphate, and magnesium derangement, metabolic acidosis, metabolic alkalosis), rhabdomyolysis, gastrointestinal bleeding, sepsis, acute myocardial infarction, ventricular arrhythmia or cardiac arrest, and end-organ failure (respiratory failure, circulatory failure, liver failure, neurological failure, hematological failure), and in-hospital mortality. Resource utilization included length of hospital stay and hospitalization cost.

### 2.4. Statistical Analysis

The total number of heatstroke patients was estimated using discharge-level weights. The clinical characteristics associated with acute kidney injury were identified using multivariable logistic regression with forward stepwise selection. The association of acute kidney injury with treatments, complications, and outcomes was evaluated using logistic regression analysis, and with length of hospital stay and hospitalization cost using linear regression analysis, with prespecified adjustment for the clinical characteristics reported in Table 1. All analyses were two-tailed. Statistical significance was achieved when *p*-value was <0.05. SPSS statistical software (version 22.0, IBM Corporation, Armonk, NY, USA) was used for all analyses.

## 3. Results

### 3.1. Incidence of and Risk Factors for Acute Kidney Injury in Hospitalized Heat Stroke Patients

There were 3372 admissions with a primary diagnosis of heat stroke during the study period. After 26 end-stage renal disease patients were excluded, a total of 3346 admissions were included in the study. Of these, 1206 (36%) had acute kidney injury, including 49 (1.5%) and 1157 (34.6%) who did and did not require dialysis in hospital, respectively. Table 1 compares the clinical characteristics of heat stroke patients with and without acute kidney injury. In a multivariable analysis, age 20–39 years, African American race, obesity, chronic kidney disease, congestive heart failure, and rhabdomyolysis were associated with increased risk of acute kidney injury. Rhabdomyolysis was the strongest risk factor for acute kidney injury. In contrast, age <20 or ≥60 years was associated with decreased risk of acute kidney injury (Table 2).

### 3.2. The Association of Acute Kidney Injury with In-Hospital Treatments, Complication, and Outcomes

Patients with acute kidney injury had greater need for invasive mechanical ventilation and blood component transfusion than patients without acute kidney injury. Acute kidney injury was significantly associated several electrolyte and acid-base derangements, sepsis, acute myocardial infarction, ventricular arrhythmia or cardiac arrest, respiratory failure, circulatory failure, liver failure, neurological failure, hematological failure, and in-hospital mortality (Table 3).

Among 49 patients with dialysis-requiring acute kidney injury, the mean age was 40 ± 16 years, and 90% were male; 45% were Caucasian, 33% were African American, and 12% were Hispanic. The in-hospital mortality was 2.7% in patients without acute kidney injury, 7.9% in patients with nondialysis-requiring acute kidney injury, and 38.8% in patients with dialysis-requiring acute kidney injury. Nondialysis-requiring and dialysis-requiring acute kidney injury was significantly associated with increased risk of in-hospital mortality, with odds ratios of 3.50 (95% CI 2.45–5.01) and 19.77 (95% CI 9.94–39.33), respectively.

### 3.3. Impact of Acute Kidney Injury on Resource Utilization

The mean length of stay in acute kidney injury patients was two days longer than for nonacute kidney injury patients. The mean hospitalization cost for patients with acute kidney injury was significantly higher than that of patients without acute kidney injury, with an additional adjusted mean of $26,974 (Table 3).

## 4. Discussion

Our large cohort of heat stroke patients demonstrated an incidence of acute kidney injury of 36%, and acute kidney injury requiring dialysis of 1.5%. Chronic kidney disease, rhabdomyolysis, obesity, male gender, congestive heart failure, young age, and African American race were factors associated with a higher risk of developing acute kidney injury. The impact of acute kidney injury on heat stroke was significant. The mortality rate in acute kidney injury patients was higher, particularly in patients requiring dialysis. Furthermore, acute kidney injury was associated with several complications, including metabolic derangements, sepsis, ventricular arrhythmia, cardiac arrest, acute myocardial infarction, and end-organ failures. Finally, acute kidney injury led to longer hospitalization and higher hospitalization costs.

Acute kidney injury is a common complication in heat stroke patients. Previous studies reported a high incidence of acute kidney injury in heat stroke, ranging from 35 to 60% [12,13,14,15]. In particular, the incidence of acute kidney injury was found to be higher in exertional heat stroke patients [6,7,16]. Furthermore, acute kidney injury was more common in heat stroke patients than in general hospitalized patients [17]. The heterogeneity of the incidence of acute kidney injury may be due to the heterogeneous populations of the studies, including the type of heat stroke, situation precipitating the heat stroke, age, sex, and race.

The impact of acute kidney injury on outcomes in heat stroke patients is considerable. It increases the mortality rate of heat stroke patients, particularly for those requiring dialysis. However, the mortality rate in renal failure patients is comparable between heat stroke and general hospitalized patients [17]. Acute kidney injury is also associated with multiple electrolyte abnormalities. Hyperkalemia is a life-threatening complication that is common in acute kidney injury. This study demonstrated that acute kidney injury significantly increased the odds of hyperkalemia. The mechanism of hyperkalemia in acute kidney injury includes the inability to excrete potassium and the release of potassium from injured muscles [18]. Other common metabolic abnormalities in acute kidney injury include hyperphosphatemia and metabolic acidosis. Although hypocalcemia is one of the most common complications of renal dysfunction, acute kidney injury was more associated with hypercalcemia in our study. Particularly with rhabdomyolysis, the release of calcium phosphate from damaged muscles during the recovery phase might be responsible for hypercalcemia [19]. Dysnatremia is another complication of renal dysfunction. While acute kidney injury is commonly associated with decreased free water excretion leading to hyponatremia, hypernatremia may also occur during the diuretic phase of acute kidney injury due to the effect of diuretic medication, or as the result of limited free water intake due to altered mental status. Our study demonstrated that acute kidney injury led to an increased risk of developing hypernatremia. A previous study reported a similar result, as well as the finding that hypernatremia in heat stroke patients was associated with higher mortality [20].

Furthermore, we found that acute kidney injury was also associated with liver failure. Kidney–liver crosstalk has been proposed as an explanation for this association. Several complications of renal dysfunction, including metabolic acidosis, azotemia, hyperphosphatemia, and inflammatory cascades, might interfere with the function of the liver and result in liver dysfunction. In addition, acute kidney injury also had crosstalk with other remote organs associated with multiorgan failures [21]. The indirect impacts of acute kidney injury on heat stroke patients include a higher risk of sepsis, which might be caused by longer hospitalization and greater need for invasive treatment. The abnormalities of electrolytes induced by renal dysfunction might precipitate cardiac arrhythmia and cardiac arrest. However, several complications, such as multiorgan failures and electrolyte abnormalities, may also be common in heat stroke. Therefore, it could not be concluded whether these complications were the result of acute kidney injury or a coexisting finding in heat stroke.

It is widely known that baseline renal function is an independent risk factor for acute kidney injury. A lower baseline eGFR was associated with a higher risk of acute kidney injury [22]. The association of rhabdomyolysis with acute kidney injury has been well recognized. Myoglobin is responsible for the main mechanisms of acute kidney injury development, including myoglobin-induced renal vasoconstriction, direct injury to tubular cells, and the formation of tubular casts [23]. A previous study also demonstrated that rhabdomyolysis increased the risk of acute kidney injury in heat stroke [6]. Furthermore, our study showed that obesity was associated with increased risk of developing acute kidney injury. Several mechanisms might explain the relationship between obesity and acute kidney injury, including glomerular hyperfiltration, a low number of functional nephrons, and the production of inflammatory cytokines from adipose tissues [24]. Male sex and black race were also previously reported as risk factors for the development of acute kidney injury [25,26]. The mechanism explaining this association remains unclear. In contrast to other studies [25,26], our study showed that younger patients (but not children) had a higher risk of developing acute kidney injury, while elderly patients had lower risk. The cause of heat stroke depends on age. We hypothesized that younger patients are more active and, subsequently, are more likely to develop exertional heat stroke than older patients. Previous studies showed that exertional heat stroke was associated with higher risk of acute kidney injury, compared to nonexertional heat stroke [6,7,16]. Therefore, younger patients are at higher risk of developing acute kidney injury. Finally, congestive heart failure was also a risk factor for acute kidney injury. Both the decreased cardiac output and the elevation of systemic venous pressure resulting from congestive heart failure led to renal hypoperfusion and acute kidney injury [27].

Some limitations of our study need to be considered. First, NIS is an inpatient-only database. Therefore, we could not evaluate long-term outcomes, such as the recovery from acute kidney injury. Second, we could not conclude whether acute kidney injury was the cause or consequence of other complications in heat stroke patients. Third, the diagnosis of acute kidney injury in this study was only based on ICD codes, and not on kidney function parameters (e.g., serum creatinine and urine output) [28]. The diagnosis of acute kidney injury on the basis of diagnosis code has low sensitivity and tends to miss mild acute kidney injury [10,11]. In addition, it did not allow us to investigate the etiology, onset, severity, and diagnostic evaluation of acute kidney injury. Finally, we did not identify the subtype, severity, and duration of heat stroke, which could have resulted in different characteristics and outcomes. To resolve these limitations, prospective study might be required to better delineate the characteristics of heat stroke and acute kidney injury. Long-term outcomes, such as, long-term mortality, recovery from acute kidney injury, dialysis dependence, and disability, should be further studied.

## 5. Conclusions

In conclusion, acute kidney injury is one of the most common complications of heat stroke, resulting in worse patient outcomes. The impacts of acute kidney injury include mortality and other complications. Baseline kidney dysfunction, obesity, male gender, congestive heart failure, young age, and African American race are factors associated with a higher risk of developing acute kidney injury.

## Figures and Tables

**Table 1 diseases-08-00028-t001:** Clinical characteristics, in-hospital treatments, complications, outcomes, and resource utilization in heat stroke patients with and without acute kidney injury.

	Total	Acute Kidney Injury	No Acute Kidney Injury	*p*-Value
Clinical characteristics				
N (%)	3346	1206 (36.0)	2140 (64.0)	
Age (years), mean (SD)	55 ± 22	50 ± 21	57 ± 23	<0.001
<20	218 (6.5)	64 (5.3)	154 (7.2)	<0.001
20–39	649 (19.4)	335 (27.8)	314 (14.7)	
40–59	1022 (30.5)	426 (35.3)	596 (27.9)	
60–79	895 (26.7)	231 (19.2)	664 (31.1)	
≥80	560 (16.7)	150 (12.4)	410 (19.2)	
Male	2459 (73.5)	985 (81.7)	1474 (69.0)	<0.001
Race				
Caucasian	1876 (56.1)	669 (55.5)	1207 (56.4)	<0.001
African American	484 (14.5)	215 (17.8)	269 (12.6)	
Hispanic	425 (12.7)	163 (13.5)	262 (12.2)	
Other	561 (16.8)	159 (13.2)	402 (18.8)	
Smoking	603 (18.0)	234 (19.4)	369 (17.2)	0.12
Alcohol drinking	270 (8.1)	97 (8.0)	173 (8.1)	0.97
Obesity	231 (6.9)	107 (8.9)	124 (5.8)	0.001
Diabetes Mellitus	549 (16.4)	153 (12.7)	369 (18.5)	<0.001
Hypertension	1233 (36.8)	409 (33.9)	824 (38.5)	0.008
Dyslipidemia	489 (14.6)	156 (12.9)	333 (15.6)	0.04
Hypothyroidism	195 (5.8)	64 (5.3)	131(6.1)	0.33
Chronic kidney disease	175 (5.2)	95 (7.9)	80 (3.7)	<0.001
Coronary artery disease	384 (11.5)	95 (7.9)	289 (13.5)	<0.001
Congestive heart failure	208 (6.2)	78 (6.5)	130 (6.1)	0.65
Atrial flutter/fibrillation	247 (7.4)	73 (6.1)	174 (8.1)	0.03
Rhabdomyolysis	1046 (31.3)	652 (54.1)	394 (18.4)	<0.001
Treatment				
Invasive mechanical ventilation	682 (20.4)	383 (31.8)	299 (14.0)	<0.001
Blood component transfusion	164 (4.9)	109 (9.0)	55 (2.6)	<0.001
Complication and outcomes				
Hyponatremia	291 (8.7)	131 (10.9)	160 (7.5)	0.001
Hypernatremia	182 (5.4)	113 (9.4)	69 (3.2)	<0.001
Hypokalemia	499 (14.9)	181 (15.0)	318 (14.9)	0.91
Hyperkalemia	132 (3.9)	100 (8.3)	32 (1.5)	<0.001
Hypocalcemia	73 (2.2)	37 (3.1)	36 (1.7)	0.008
Hypercalcemia	37 (1.1)	29 (2.4)	8 (0.4)	<0.001
Hypo/hyperphosphatemia	120 (3.6)	72 (6.0)	48 (2.2)	<0.001
Hypo/hypermagnesemia	128 (3.8)	58 (4.8)	70 (3.3)	0.03
Metabolic acidosis	468 (14.0)	296 (24.5)	172 (8.0)	<0.001
Metabolic alkalosis	29 (0.9)	18 (1.5)	11 (0.5)	0.003
Gastrointestinal bleeding	53 (1.6)	30 (2.5)	23 (1.1)	0.002
Sepsis	144 (4.3)	93 (7.7)	51 (2.4)	0.001
Acute myocardial infarction	221 (6.6)	100 (8.3)	121 (5.7)	0.003
Ventricular arrhythmia/cardiac arrest	94 (2.8)	57 (4.7)	37 (1.7)	<0.001
Respiratory failure	546 (16.3)	324 (26.9)	222 (10.4)	<0.001
Circulatory failure	385 (11.5)	199 (16.5)	186 (8.7)	<0.001
Liver failure	192 (5.7)	147 (12.2)	45 (2.1)	<0.001
Neurological failure	641 (19.2)	305 (25.3)	336 (15.7)	<0.001
Hematological failure	445 (13.3)	272 (22.6)	173 (8.1)	<0.001
In-hospital mortality	167 (5.0)	110 (9.1)	57 (2.7)	<0.001
Resource utilization				
Length of hospital stay (days), mean (SD)	4.2 ± 6.7	5.5±8.8	3.5 ± 5.0	<0.001
Hospitalization cost ($), mean (SD)	34,780 ± 70,401	52,656 ± 103,477	24,707 ± 37,912	<0.001

**Table 2 diseases-08-00028-t002:** Factors associated with acute kidney injury in heat stroke patients.

Variables	Univariable Analysis		Multivariable Analysis	
	Crude Odds Ratio (95%CI)	*p* Value	Adjusted Odds Ratio (95%CI)	*p* Value
Age (years)				
<20	0.58 (0.42–0.80)	0.001	0.60 (0.43–0.83)	0.002
20-39	1.49 (1.23–1.82)	<0.001	1.47 (1.20–1.80)	<0.001
40-59	1 (reference)		1 (reference)	
60-79	0.49 (0.40–0.59)	<0.001	0.54 (0.44–0.66)	<0.001
≥80	0.51 (0.41–0.64)	<0.001	0.57 (0.44–0.72)	<0.001
Male	2.01 (1.69–2.39)	<0.001	1.67 (1.39–2.01)	<0.001
Race				
Caucasian	1 (reference)		1 (reference)	
African American	1.44 (1.18–1.77)	<0.001	1.34 (1.09–1.66)	0.006
Hispanic	1.12 (0.90–1.40)	0.30	0.96 (0.77–1.21)	0.74
Other	0.71 (0.58–0.88)	0.001	0.71 (0.57–0.88)	0.002
Smoking	1.16 (0.96–1.39)	0.12		
Alcohol drinking	0.99 (0.77–1.29)	0.97		
Obesity	1.58 (1.21–2.07)	0.001	1.62 (1.21–2.16)	0.001
Diabetes Mellitus	0.64 (0.52–0.78)	<0.001		
Hypertension	0.82 (0.71–0.95)	0.008		
Dyslipidemia	0.81 (0.66–0.99)	0.04		
Hypothyroidism	0.86 (0.63–1.17)	0.34		
Chronic kidney disease	2.20 (1.62–2.99)	<0.001	3.14 (2.27–4.36)	<0.001
Coronary artery disease	0.55 (0.43–0.70)	<0.001		
Congestive heart failure	1.07 (0.80–1.43)	0.65	1.48 (1.08–2.04)	0.02
Atrial flutter/fibrillation	0.73 (0.55–0.97)	0.03		
Rhabdomyolysis	5.22 (4.46–6.10)	<0.001	4.64 (3.92–5.49)	<0.001

**Table 3 diseases-08-00028-t003:** The association between acute kidney injury and in-hospital treatment, complications, outcomes, and resource utilization in heat stroke patients.

	Univariable Analysis		Multivariable Analysis	
	Crude Odds Ratio (95% CI)	*p* Value	Adjusted Odds Ratio * (95% CI)	*p* Value
Treatments				
Invasive mechanical ventilation	2.87 (2.41–3.40)	<0.001	2.84 (2.35–3.42)	<0.001
Blood component transfusion	3.77 (2.70–5.25)	<0.001	3.62 (2.55–5.14)	<0.001
Complications and outcomes				
Hyponatremia	1.51 (1.18–1.92)	0.001	1.56 (1.21–2.02)	0.001
Hypernatremia	3.10 (2.28–4.22)	<0.001	3.86 (2.79–5.36)	<0.001
Hypokalemia	1.01 (0.83–1.23)	0.91	1.04 (0.85–1.29)	0.69
Hyperkalemia	5.96 (3.97–8.93)	<0.001	5.51 (3.62–8.40)	<0.001
Hypocalcemia	1.85 (1.16–2.94)	0.009	1.75 (1.07–2.86)	0.03
Hypercalcemia	6.57 (2.99–14.41)	<0.001	5.28 (2.34–11.92)	<0.001
Hypo/hyperphosphatemia	2.77 (1.91–4.02)	<0.001	2.57 (1.74–3.80)	<0.001
Hypo/hypermagnesemia	1.49 (1.05–2.13)	0.03	1.52 (1.05–2.22)	0.03
Metabolic acidosis	3.72 (3.04–4.56)	<0.001	3.72 (3.00–4.62)	<0.001
Metabolic alkalosis	2.93 (1.38–6.23)	0.005	3.06 (1.38–6.80)	0.006
Gastrointestinal bleeding	2.35 (1.36–4.06)	0.002	1.05 (0.49–2.26)	0.89
Sepsis	3.42 (2.41–4.85)	<0.001	3.63 (2.51–5.25)	<0.001
Acute myocardial infarction	1.51 (1.15–1.99)	0.003	1.86 (1.39–2.51)	<0.001
Ventricular arrhythmia/cardiac arrest	2.82 (1.85–4.29)	<0.001	2.89 (1.86–4.52)	<0.001
Respiratory failure	3.17 (2.63–3.83)	<0.001	3.26 (2.66–3.98)	<0.001
Circulatory failure	2.08 (1.68–2.57)	<0.001	2.10 (1.67–2.63)	<0.001
Liver failure	6.46 (4.59–9.10)	<0.001	5.62 (3.92–8.04)	<0.001
Neurological failure	1.82 (1.53–2.16)	<0.001	1.94 (1.61–2.33)	<0.001
Hematological failure	3.31 (2.70–4.07)	<0.001	3.01 (2.42–3.75)	<0.001
In-hospital mortality	3.67 (2.65–5.10)	<0.001	3.95 (2.79–5.59)	<0.001
	**Coefficient (95% CI)**	***p*** **Value**	**Adjusted Coefficient* (95% CI)**	***p*** **Value**
Resource utilization				
Length of hospital stay (days)	2.0 (1.5–2.5)	<0.001	2.0 (1.6–2.5)	<0.001
Hospitalization cost ($)	27949 (23,035–32,863)	<0.001	26974 (21,933–32,014)	<0.001

* Adjusted for age, sex, race, smoking, alcohol drinking, obesity, diabetes mellitus, hypertension, dyslipidemia, hypothyroidism, chronic kidney disease, chronic ischemic heart disease, congestive heart failure, and atrial flutter/fibrillation.

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
