# Peer review of "Impact of Acute Kidney Injury on Outcomes of Hospitalizations for Heat Stroke in the United States"

_diseases, 2020, doi:10.3390/diseases8030028_

Round 1
Reviewer 1 Report
The overall topic is interesting and relevant . This single center study registers heat stroke patients and the occurrence of acute kidney failure and other changes such as electrolyte changes. Interestingly, young people were significantly affected.
The study presents many data, however some questions and items should be addressed
-A problem is that the diagnosis of kidney failure is only based on the ICD codes, and not on given kidney function parameters. moreover, it is not clear when acute kidney failure started to develop; this must be addressed in the discussion section.
-Using e.g. the ICD code 584.7, how was “medullary renal damage” diagnosed, if no biopsy was performed? Authors should comment on that.
-In the results section, the description of the results in the text (already shown in tables) is sometimes somewhat redundant, and should be focused.
-In the abstract, the listing of each individual electrolyte is overstated.
-In the discussion section authors should comment on how prospective studies should be conceptualized, and how patient with ANV and heat stroke patients should be tracked nephrologically.
-How many patients remained on dialysis therapy in the long run ?
-There were also patients with acute on chronic renal failure, how was this defined?
-Why were younger people more often affected by acute kidney injury in this analysis?
-How was heat stroke patients in generally treated?
-In the analysis, rhabdomyolysis should be more emphazised as a critical step towards acute renal failure.
Author Response
Response to Reviewer #1
The overall topic is interesting and relevant. This single center study registers heat stroke patients and the occurrence of acute kidney failure and other changes such as electrolyte changes. Interestingly, young people were significantly affected.
The study presents many data, however some questions and items should be addressed
Response: We thank you for reviewing our manuscript and for your critical evaluation.
Comment #1
A problem is that the diagnosis of kidney failure is only based on the ICD codes, and not on given kidney function parameters. moreover, it is not clear when acute kidney failure started to develop; this must be addressed in the discussion section.
Response: The reviewer raises important point. The following statements have been added to the limitation to address the problem of the diagnosis of acute kidney injury based on ICD code
“Third, the diagnosis of acute kidney injury in this study was only based on ICD codes, and not on kidney function parameters (e.g. serum creatinine and urine output). The diagnosis of acute kidney injury on the basis of diagnosis code has low sensitivity and tends to miss mild acute kidney injury (6, 7). In addition, it does not allow us to investigate the etiology, onset, severity and diagnostic evaluation of acute kidney injury.”
Comment #2
Using e.g. the ICD code 584.7, how was “medullary renal damage” diagnosed, if no biopsy was performed? Authors should comment on that.
Response: We appreciate the reviewer input. We agree. Please see the response to comment 1
Comment #3
In the results section, the description of the results in the text (already shown in tables) is sometimes somewhat redundant, and should be focused.
Response: We agree with the reviewer. This has been revised as suggested.
Comment #4
In the abstract, the listing of each individual electrolyte is overstated.
Response: We agree with the reviewer. This has been revised as suggested.
Comment #5
In the discussion section authors should comment on how prospective studies should be conceptualized, and how patient with AKI and heat stroke patients should be tracked nephrologically.
Response: The reviewer raises an important point. We agree and the following statements have been added to discussion.
“To resolve these limitations, prospective study might be required to better delineate characteristics of heat stroke and acute kidney injury. Long-term outcomes, such as, long-term mortality, recovery from acute kidney injury, dialysis dependence, and disability, should be further studied.”
Comment #6
How many patients remained on dialysis therapy in the long run?
Response: We appreciate the reviewer’s input. As the National Inpatient Sample contains only hospitalization data, we did not have data on outcomes after hospital discharge. The following statements have been added to the limitation.
“NIS is an inpatient-only database. Therefore, we could not evaluate long-term outcomes, such as the recovery from acute kidney injury.”
Comment #7
There were also patients with acute on chronic renal failure, how was this defined?
Response: Thank you for important comment. Chronic kidney disease was defined based on ICD-9 diagnosis code of 585.1 (chronic kidney disease stage 1), 585.2 (chronic kidney disease stage 2), 585.3 (chronic kidney disease stage 3), 585.4 (chronic kidney disease stage 4), and 585.9 (chronic kidney disease, unspecified). Patients with end-stage renal disease (585.6) and chronic kidney disease stage 5 (585.5) were excluded from this study.
Comment #8
Why were younger people more often affected by acute kidney injury in this analysis?
Response: The following statements have been added to the discussion to explain the possible reason why younger patients had more acute kidney injury.
“The cause of heat stroke depends on age. We hypothesized that younger patients are more active and, subsequently, are more likely to develop exertional heat stroke than older patients. Previous studies showed exertional heat stroke was associated with higher risk of acute kidney injury, compared to non-exertional heat stroke. Therefore, younger patients are at higher risk of developing acute kidney injury.”
Comment #9
How was heat stroke patients in generally treated?
Response: The following statements have been added to introduction to describe the general treatment of heat stroke.
“Rapid and effective cooling along with supportive treatment are fundamental treatment of heat stroke. The targets of cooling are to reduce the body core temperatures to 39 oC within 10-40 minutes and to 38.5 oC or below within 2 hours (3). Although both invasive and noninvasive method has been used in clinical practices, there has been no prospective study demonstrating the superiority of whether cooling method. Supportive treatment, including fluid resuscitation, electrolyte abnormalities correction, and organ failure supports (2).”
Comment #10
In the analysis, rhabdomyolysis should be more emphazised as a critical step towards acute renal failure.
Response: We appreciate the reviewer’s important comment. The following statements have been added to result.
“In multivariable analysis, age 20-39 years, African American race, obesity, chronic kidney disease, congestive heart failure, and rhabdomyolysis were associated with increased risk of acute kidney injury. Rhabdomyolysis was the strongest risk factors for acute kidney injury.”
The following statements have been added to discussion
“The association of rhabdomyolysis with acute kidney injury has been well recognized. Myoglobin is responsible for the main mechanisms of acute kidney injury development, including myoglobin induced renal vasoconstriction, direct injury to tubular cells, and the formation of tubular casts (20). The previous study also demonstrated that rhabdomyolysis increased the risk of acute kidney injury in heat stroke (4).”
We greatly appreciated the editor and reviewer’s time and comments to improve our manuscript.

Reviewer 2 Report
I am happy to review this paper, “Impact of Acute Kidney Injury on Outcomes of Hospitalizations for Heat Stroke in the United States”. Authors were to evaluate the risk factors and the association of acute kidney injury with treatments, complications, outcomes, and resource utilization in hospitalized patients for heat stroke. They found that acute kidney injury in heat stroke patients was associated with mortality and other complications. They suggest that baseline kidney dysfunction, obesity, male gender, congestive heart failure, young age, and African American race are factors associated with a higher risk of developing acute kidney injury. This is well written nice paper but I think some issues need to be resolved.
- Need to adjust factors with the severity of heat stoke
It’s hard to understand that patients with elder, DM, hypertension, coronary artery disease, dyslipidemia are less associated with AKI in heat stroke. It will be discussed in the manuscript. Data about heat stroke severity such as duration, initial temperature, mental status, and initial hypotension need to be adjusted in multivariable analysis.
- Reliability of AKI code
I’m concern about the reliability of AKI such a long study period. In that period, physicians may be used the concept of acute renal failure and acute kidney injury. Moreover someone defined AKI by RIFLE, AKIN, or KDIGO.
3. The characteristics of patients with AKI requiring dialysis will be helpful information
Author Response
Response to Reviewer #2
I am happy to review this paper, “Impact of Acute Kidney Injury on Outcomes of Hospitalizations for Heat Stroke in the United States”. Authors were to evaluate the risk factors and the association of acute kidney injury with treatments, complications, outcomes, and resource utilization in hospitalized patients for heat stroke. They found that acute kidney injury in heat stroke patients was associated with mortality and other complications. They suggest that baseline kidney dysfunction, obesity, male gender, congestive heart failure, young age, and African American race are factors associated with a higher risk of developing acute kidney injury. This is well written nice paper but I think some issues need to be resolved.
Response: We thank you for reviewing our manuscript and for your critical evaluation.
Comment #1
Need to adjust factors with the severity of heat stoke
It’s hard to understand that patients with elder, DM, hypertension, coronary artery disease, dyslipidemia are less associated with AKI in heat stroke. It will be discussed in the manuscript. Data about heat stroke severity such as duration, initial temperature, mental status, and initial hypotension need to be adjusted in multivariable analysis.
Response: We agree with your suggestion that we should have included severity of heat stroke in adjusting factors. Unfortunately, the data on severity of heat stroke was lacking in the database. The following statements have been included in the limitation section.
“We did not identify the subtype, severity, and duration of heat stroke, which could have resulted in different characteristics and outcomes.”
Of note, diabetes mellitus, hypertension, coronary artery disease and dyslipidemia were associated with less acute kidney injury only in univariate analysis but these factors were not significantly associated with acute kidney injury in the multivariate analysis.
Comment #2
Reliability of AKI code
I’m concern about the reliability of AKI such a long study period. In that period, physicians may be used the concept of acute renal failure and acute kidney injury. Moreover someone defined AKI by RIFLE, AKIN, or KDIGO.
Response: The following statements have been included in the data collection section to describe the accuracy of AKI diagnosis based on ICD code
“The identification of acute kidney injury using ICD-9 diagnosis code has a specificity of 98% but a sensitivity of 17% and is likely to capture a more severe spectrum of acute kidney injury, compared with KDIGO serum creatinine-based criteria.”
The following statements have been added to the limitation to address the problem of the diagnosis of acute kidney injury based on ICD code.
“The diagnosis of acute kidney injury in this study was only based on ICD codes, and not on kidney function parameters (e.g. serum creatinine and urine output). The diagnosis of acute kidney injury on the basis of diagnosis code has low sensitivity and tends to miss mild acute kidney injury. In addition, it does not allow us to investigate the etiology, onset, severity, and diagnostic evaluation of acute kidney injury.”
Reference
Grams ME, Waikar SS, MacMahon B, Whelton S, Ballew SH, Coresh J: Performance and limitations of administrative data in the identification of AKI. Clin J Am Soc Nephrol, 9: 682-689, 2014
Waikar SS, Wald R, Chertow GM, Curhan GC, Winkelmayer WC, Liangos O, Sosa MA, Jaber BL: Validity of International Classification of Diseases, Ninth Revision, Clinical Modification Codes for Acute Renal Failure. J Am Soc Nephrol, 17: 1688-1694, 2006
Comment #3
The characteristics of patients with AKI requiring dialysis will be helpful information
Response: The reviewer raises important point. The following statements have been added to result
“Among 49 patients with dialysis-requiring acute kidney injury, the mean age was 40±16 years and 90% were male. 45% were Caucasian, 33% were African American, and 12% were Hispanic.”
We greatly appreciated the editor and reviewer’s time and comments to improve our manuscript.

Round 2
Reviewer 2 Report
I have no further comments and congratulate on your research.